# Long-Term Operation of a Pilot-Scale Sulfur-Based Autotrophic Denitrification System for Deep Nitrogen Removal

**Yan Wang [1], Weiyi Xu [1], Xue Yang [1], Zhengming Ren [1], Kaiwen Huang [1], Feiyue Qian [2,3] and Ji Li [1,3,*]**

[1] Jiangsu Key Laboratory of Anaerobic Biotechnology, School of Environment and Civil Engineering, Jiangnan University, Wuxi 214122, China
[2] School of Environmental Science and Engineering, Suzhou University of Science and Technology, Suzhou 215009, China
[3] Jiangsu College of Water Treatment Technology and Material Collaborative Innovation Center, Suzhou 215009, China
[*] Correspondence: liji@jiangnan.edu.cn

**Abstract:** Sulfur-based autotrophic denitrification is a novel biological denitrification process characterized by the absence of an organic carbon source, a short reaction time, a high denitrification rate, a low treatment cost, and a small footprint. However, the technique is facing challenges with respect to engineering applications. In this study, a pilot-scale sulfur-based autotrophic denitrification system was established with an optimal hydraulic retention time (HRT) of 0.21 h, which achieved the highest denitrification load of 1158 mg/(L·d) and a denitrification rate of 164 g$NO_3^-$-N/(m$^3$·h). Effective backwashing is the basis for the long-term stable and efficient nitrogen removal performance, which recovered its normal nitrogen removal performance within 0.5 h. In addition, the operation cost is merely 0.013 $/t, indicating that the sulfur-based autotrophic denitrification process presents good economic applicability, and the relatively low operation cost will lay a foundation for practical application.

**Keywords:** sulfur-based autotrophic denitrification; pilot-scale bioreactor; long-term operation and regulation; microbial community structure; cost analysis





## 1. Introduction

Nitrogen and phosphorus pollutants are critical factors for eutrophication in water bodies. At present, more than 56% major lakes in China are in the state of eutrophication due to nitrogen and phosphorus pollution. While the effluent from most urban wastewater treatment plants (WWTPs) has met the Grade A criteria specified in the Discharge Standard of Pollutants for Municipal Wastewater Treatment Plant (GB 18918-2002), nitrogen and phosphorus in the effluent affect ecologically sensitive areas; therefore, more stringent discharge standards are subsequently issued, where total nitrogen (TN) and total phosphorus (TP) in the effluent should be lower than 10 mg/L and 0.3 mg/L [1,2], respectively. It is necessary to achieve a further significant reduction in TN and TP concentrations in effluents, even if the effluent quality meets Grade A standards (TN $\leq$ 15 mg/L and TP $\leq$ 0.3 mg/L).

The mainstream nitrogen removal technology is principally the biochemical method, while physical and chemical approaches involve high operating costs and are, therefore, less frequently employed in domestic sewage treatment processes. Nitrogen removal generally refers to ammonification, nitrification, and denitrification by microorganisms, comprising autotrophic and heterotrophic denitrification. Heterotrophic denitrification denotes that denitrifying bacteria reduce nitrate ($NO_3^-$-N) and nitrite ($NO_2^-$-N) to nitrogen ($N_2$) in an anoxic environment, with organic carbon as the energy source, thus entailing a high carbon-to-nitrogen ratio (C/N) in the influent. Due to the influence of dissolved oxygen

(DO) and the biological phosphorus removal process, C/N is generally controlled at four to five to achieve a relatively high total nitrogen removal.

To further reduce the effluent nitrate, additional carbon sources are needed, which, however, increases treatment costs and makes it more difficult to control the dosage, potentially causing secondary pollution. Hence, sulfur-based autotrophic denitrification without organic carbon sources is gradually becoming a research hotspot, and its specific nitrogen removal process is shown in Figure 1. The sulfur-based autotrophic denitrification process is characterized by a high nitrogen removal efficiency and a short hydraulic reaction time; conversely, there is a certain demand for alkalinity and sulfate will be generated as a by-product. The consumption of alkalinity and the production of sulfate are related to the influent nitrate concentration. Consequently, the process is suitable for treating low-concentration nitrate wastewater when there is a limit to the effluent sulfate concentration and no alkalinity is added. This indicates that the approach fits WWTP with a low C/N ratio in the secondary effluent mainly containing $NO_3^-$-N. While many studies have focused on the application of the sulfur-based autotrophic denitrification process to groundwater and drinking water denitrification [3,4], further investigation and in-depth research of wastewater denitrification with respect to operational and cost control, reduction of secondary hazards, and elimination of ecological risks is warranted.

$$55S + 50NO_3^- + 32H_2O + 20CO_2 + 4NH_4^+ \xrightarrow{\text{thiobacillus denitrificans}} 4C_5H_7O_2N + 55SO_4^{2-} + 25N_2 + 64H^+$$

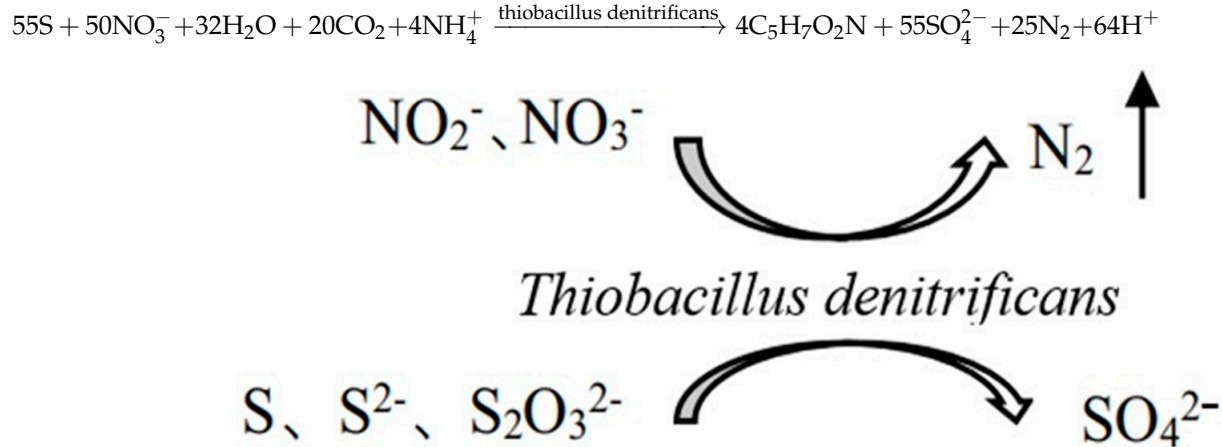

**Figure 1.** Sulfur autotrophic denitrification process.

In this paper, a pilot-scale bioreactor was established to analyze the start-up, long-term stable operation, and backwash recovery process. Nitrogen removal performance of the pilot-scale bioreactor under different flow rates was investigated, and the dissolved oxygen (DO) and nitrate concentration were measured to calculate the nitrogen removal rate. The limit treatment capacity of the pilot-scale bioreactor was explored by monitoring pH variations and long-term operation performance. In addition, the microbial community structure at different heights of pilot-scale bioreactor using bacterial 16S rRNA cloning library were analyzed. The microbial community structure was further developed to reveal the discrepancy of the lab-scale and pilot-scale bioreactor. Finally, the operation costs of the pilot-scale bioreactor were estimated to verify the economic feasibility for practical application.

## 2. Materials and Methods

### 2.1. Establishment of Pilot-Scale Bioreactor

The pilot-scale bioreactor was a column-shaped vessel (Figure 2) made of stainless steel, with a height of 3.4 m (effective height of 3.1 m), an inner diameter of 1.1 m, and an effective volume of 2.94 m³. The section of 0.5 m height at the bottom of the pilot-scale bioreactor is the influent buffer zone, and the upper part of the buffer zone is equipped with a water distributor. The upper part of the water distributor is paved with coarse sand (with a particle size of 4~8 mm) at a height of 0.1 m. In addition, the upper 1.8 m of the

coarse sand is filled with sulfur particles with a total mass of approximately 1.75 t and a porosity of approximately 50%. Sampling ports were provided at 0.6, 1.0, 1.5, 2.0, and 2.5 m from the bottom of the pilot-scale bioreactor. Furthermore, a pressure gauge was installed at the sampling port 0.6 m from the bottom to monitor pressure changes, so as to characterize blockage of sulfur particles inside the pilot-scale bioreactor. In addition, a lab-scale bioreactor was established to simultaneously evaluate the nitrogen removal performance of the pilot-scale bioreactor, with the same operation parameters. The effective volume was 8.0 L (inner diameter: 20 cm; height: 35 cm), and the bioreactor was filled with the coarse sand and sulfur particles.

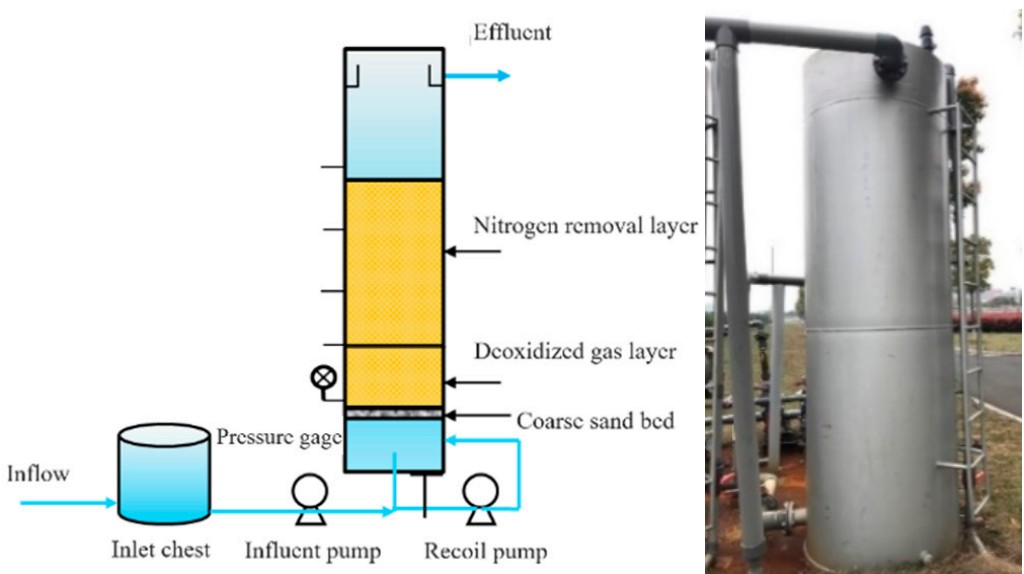

**Figure 2.** Schematic diagram of the pilot-scale bioreactor.

### 2.2. Sulfur Particles

Sulfur particles with different particle sizes show great impact on the start-up and treatment capacity of the pilot-scale bioreactor. Small sulfur particles have a large specific surface area and are, thus, efficient in treatment [5]; however, the small size sulfur particles tend to flow out and limit the practical applications. Therefore, sulfur particles were divided into hemispherical flakes, with a particle size of 4–8 mm, and large granules, with a particle size of approximately 1 cm.

At a backwashing intensity of 38 L/(m²-s), sulfur particles with a size of 1 cm could not be fluidized, while hemispherical sulfur particles with a size of 4–8 mm could be fluidized at a backwashing intensity of 19 L/(m²-s). Thus, when using sulfur with a particle size of 1 cm, a large backwashing flow rate is required to achieve fluidization. However, an excessive flow rate may scour off the biofilm from the sulfur particles; therefore, hemispherical industrial sulfur with a particle size of 4–8 mm was adopted.

### 2.3. Seed Sludge and Influent

The pilot-scale bioreactor was supplied with the seed sludge from the thickening tank in Xinjian (Yixing, China) WWTP through an anaerobic–anoxic–oxic process, and the influent was introduced from the effluent of the secondary sedimentation tank. At the start-up stage, potassium nitrate was added to the influent to increase the nitrate loading, and the effluent quality of the secondary sedimentation tank is shown in Table 1.

**Table 1.** Effluent quality of the secondary sedimentation tank.

| COD (mg/L) | NH$_4^+$-N (mg/L) | NO$_3^-$-N (mg/L) | TP (mg/L) | pH | T (°C) |
|---|---|---|---|---|---|
| 30~50 | 0.1~0.5 | 6~15 | 0.2~0.5 | 6.9~7.3 | 13~27 |

*2.4. Analytical Methods*

Ammonia nitrogen, nitrate, total nitrogen (TN), mixed liquor suspended solids (MLSS), and mixed liquor volatile suspended solids (MLVSS) were measured according to the standard method [6]. To investigate the succession of functional microorganisms in the pilot-scale bioreactor, the biofilm was periodically sampled and the DNA from biofilms was extracted using a soil bacterial genomic DNA extraction kit. The DNA was then sequenced by Illumina Miseq after PCR amplification and product purification, for which the sequencing library was established and completed by Genergy Bio-Technology (Shanghai, China). In addition, the nitrogen removal rate (g NO$_3^-$- N/m$^3$·h) was expressed as

$$v = \frac{(C_i - C_e)Q}{V\rho} \tag{1}$$

where C$_i$ and C$_e$ denote the nitrate in the influent and effluent (mg/L), respectively; Q represents the influent flow rate (m$^3$/h); V and ρ define the volume (m$^3$) the porosity (%) of the sulfur particles.

**3. Results and Discussion**

*3.1. Start-Up of the Pilot-Scale Bioreactor*

The sulfur-based autotrophic denitrification biofilm was cultivated in recirculating influent and effluent mode, and the start-up of the pilot-scale bioreactor was initiated in two stages. Stage I (0–30 d), the sludge from the thickening tank was inoculated in the influent tank with a dosage of 50 L/d, then the sludge mixture was pumped into the pilot-scale bioreactor with the flow rate of 1 m$^3$/h. Finally, the effluent was introduced to the pilot-scale bioreactor to enhance biofilm formation. Stage II (31–130 d), sludge inoculation was terminated and the influent nitrate was maintained up to 30 mg/L by adding KNO$_3$, which is beneficial to the growth of autotrophic denitrification biofilm. Moreover, the pilot-scale bioreactor ought to be periodically and completely discharged to prevent excessive accumulation of toxic substances from microorganisms, whilst the influent flow rate was kept at 1 m$^3$/h. Since the formation of autotrophic denitrification biofilm is relatively slow, it took a long time to achieve the successful start-up of the pilot-scale bioreactor in winter. At the beginning of the start-up period, the maximum influent temperature was only 18 °C, which was not conducive to microbial growth [7]; therefore, a was set up between the influent tank and the influent pump to maintain the influent temperature at the optimum of 30 °C.

As illustrated in Figure 3, the denitrification performance was bad in the middle start-up period, and the respective effluent nitrate and denitrification load was approximately 10 mg/L and 35–40 mg/(L·d) after 4 d of cycling. Thus, cycling cultivation mode was still required. However, the operation mode was changed to low-flow continuous influent and effluent when the start-up period of the pilot-scale bioreactor exceeded for 100 d, under which the treatment capacity gradually improved for approximately 25 days, and the effluent nitrate ranged from 0.1 to 2.4 mg/L (Figure 4). At 127 d, the influent nitrate was 12 mg/L, the nitrate removal rate was higher than 99% and the denitrification load was up to 97 mg/(L·d). Brown biofilm was observed and attached to the sulfur particles (Figure 5), implying that the start-up of the pilot-scale bioreactor was successful.

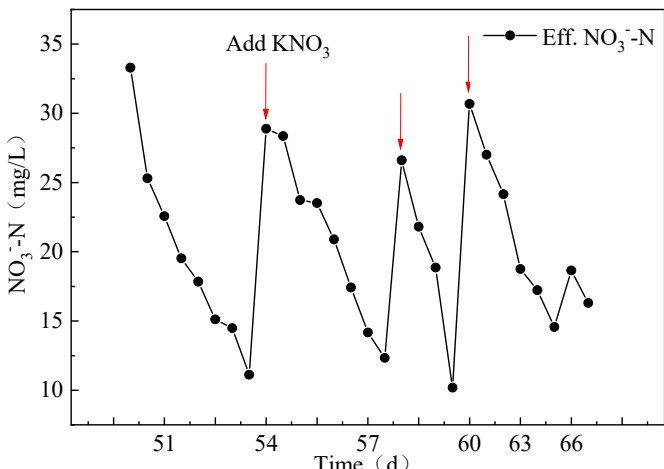

**Figure 3.** Effluent nitrate variation during the start-up process of the pilot-scale bioreactor.

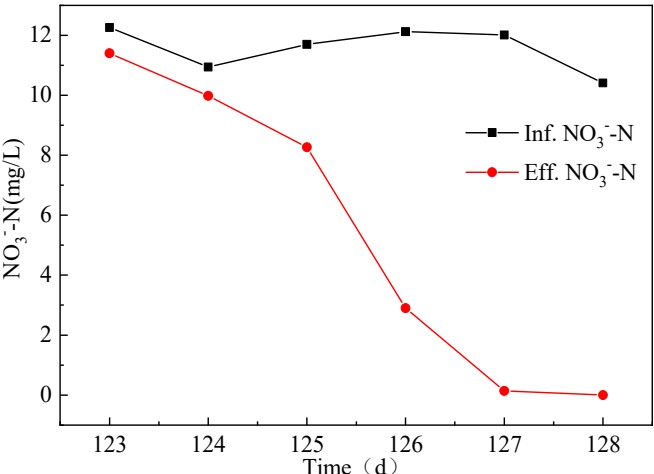

**Figure 4.** Effluent nitrate variation during the start-up process of the pilot-scale bioreactor by the low-flow effluent strategy.

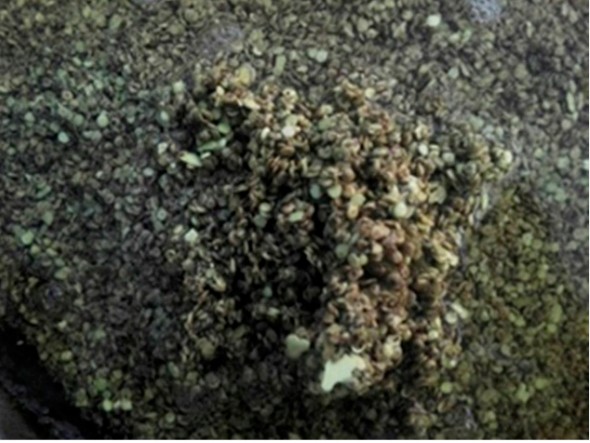

**Figure 5.** Sulfur particles with biofilm attached.

### 3.2. Nitrate Removal under Different HRT Conditions

After the start-up of the pilot-scale bioreactor, the influent flow rate was set to 2.0, 4.0, 6.0, 8.0, 10.0, 12.0, 14.0, and 16.0 $m^3$/h to gradually increase the denitrification load, and the corresponding HRT was 1.47, 0.74, 0.49, 0.37, 0.29, 0.24, 0.21, and 0.18 h, respectively. Nitrate removal was stably higher than 60% when HRT was less than 0.21 h (Figure 6),

and average effluent nitrate was lower than 1.0 mg/L when HRT ranged between 1.47 and 0.74 h, and nitrate removal was as high as 90%; conversely, the denitrification was 223 mg/ (L·d) (Figure 7). When the effluent nitrate was stable below 6.0 mg/L, the performance of increasing denitrification load and treatment capacity was evaluated by gradually increasing the influent flow rate. The pilot-scale bioreactor demonstrated high denitrification properties with an average efficiency of 84% and an increased average denitrification load of 414 mg/(L·d) under the HRT condition ranged between 0.49 and 0.37 h. In addition, when the HRT was reduced to 0.29–0.24 h, nitrate removal decreased to approximately 75%, while the effluent nitrate was still lower than 4 mg/L, and the denitrification load was further increased to 558 mg/(L·d). The average effluent nitrate was below 3.6 mg/L even when HRT was 0.21 h, and the denitrification load increased to 902 mg/(L·d) and maximized to 1158 mg/(L·d). However, when the HRT was further reduced to 0.18 h, the influent flow rate reached 17.2 m$^3$/h, and effluent nitrate exceeded 6.0 mg/L, the corresponding nitrate removal rate decreased to 33%, and the denitrification load reduced to 382 mg/(L·d). Therefore, based on the nitrogen removal performance and treatment capacity of the pilot-scale bioreactor, the subsequent stable operation flow rate is set to 14.0 m$^3$/h, HRT is 0.21 h, and the maximum treatment capacity is 336 m$^3$/d.

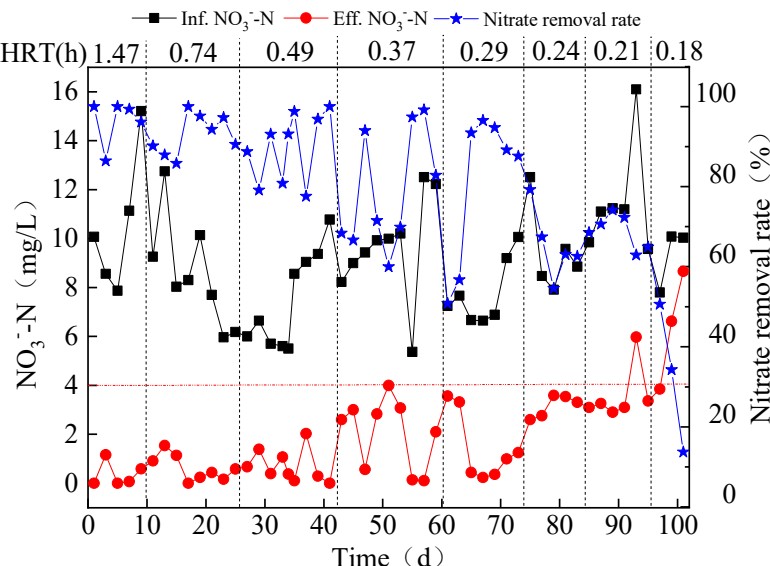

**Figure 6.** Nitrate removal performance under different HRT.

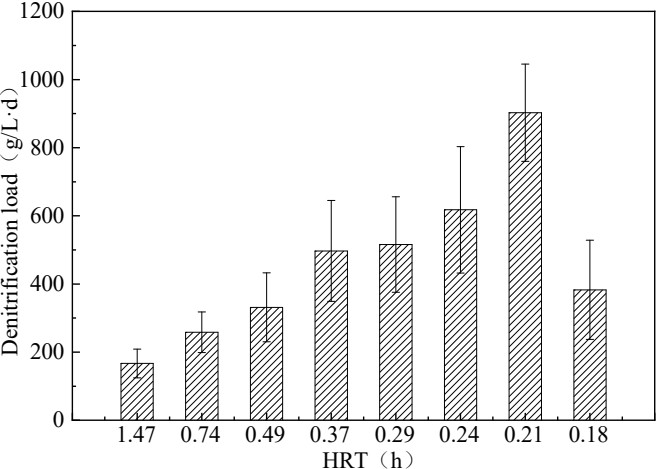

**Figure 7.** Nitrogen removal loading under different HRT.

The maximum denitrification load of 1750 mg/(L·d) was achieved in a lab-scale bioreactor [8], while the pilot-scale bioreactor reached 1158 mg/(L·d). The possible reasons could be due to (1) the influent of the lab-scale bioreactor being supplied with the synthetic sewage, so the microbial activity of biofilm can be well maintained; (2) the denitrification load being associated with the nitrate concentration. The influent nitrate of the lab-scale bioreactor was of 30 mg/L, which is significantly higher than that of the pilot-scale bioreactor (5–10 mg/L), thus limiting the denitrification load.

### 3.3. Variation of Nitrate and DO at Different Heights in the Pilot-Scale Bioreactor

The sampling points are set to 0.6, 1.0, 1.5, 2.0, and 2.5 m away from the bottom and the effluent outlet of the pilot-scale bioreactor, respectively, to monitor the nitrate and DO concentrations at the influent flow rate of 2.0, 6.0, 10.0, 12.0, and 14.0 m$^3$/h. The functional flora of sulfur-based autotrophic denitrification process is facultative microorganism. Under high DO conditions, the strains oxidized sulfur to obtain energy and thus consumed oxygen [9]; however, it performed autotrophic denitrification capability under low DO conditions. Therefore, in order to achieve a high nitrate removal performance, it is necessary to remove DO of the secondary effluent and, thus, create a good denitrification environment for autotrophic denitrifying bacteria.

As shown in Figure 8A, a close correlation was discovered between nitrate concentration in the denitrification layer (1.1–2.4 m) and the height of the sulfur particles' bed. The total height of the sulfur particles' bed in the pilot-scale bioreactor was 1.8 m, of which 0.5 m at the bottom were used for DO removal in the influent, and the remaining 1.3 m were used for nitrate removal. The filling porosity of the sulfur particles' bed was approximately 50%, and consequently, the volume was 0.617 m$^3$ when the height of the sulfur particles' bed was 1.3 m, and the average nitrogen removal rate of the pilot-scale bioreactor was 164 gNO$_3^-$-N/(m$^3$·h).

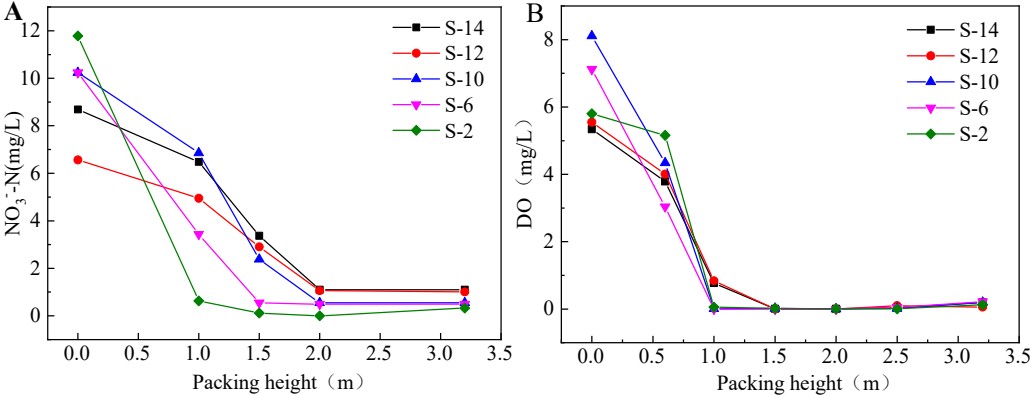

**Figure 8.** Nitrate (**A**) and DO (**B**) concentration variations in the pilot-scale bioreactor.

Figure 8B illustrates that the DO concentration in the deoxygenation layer (1.1 m) approximated zero under the condition of low flow rate at 2.0–6.0 m$^3$/h (HRT 1.47~0.49 h). Nitrate concentration substantially decreased, mainly because the flow rate was low, the HRT was long, and there was sufficient time for denitrification after DO removal in the deoxidization layer. When the flow rate was elevated and the HRT was 0.37–0.21 h, the DO concentration increased to 0.8 mg/L because deoxygenation was insufficient for denitrification. The possible reason could be attributed to a lower denitrification efficiency as a result of an increased flow rate and a shortened HRT. In addition, the influent DO of the pilot-scale bioreactor experienced a slow increase after several water drop process [10] when the influent flowed from the biochemical and secondary sedimentation tanks. Therefore, to improve the denitrification efficiency, the influent DO ought to be reduced as much as possible to create a suitable nitrate removal environment in the denitrification layer.

### 3.4. Long-Term Stable Operation of the Pilot-Scale Bioreactor

The influent flow was set at 14 m$^3$/h, HRT at 0.21 h, and the operation period lasted for about 120 d. Additionally, the sewage temperature was in the range of 13–27 °C.

#### 3.4.1. Variation of Nitrate

The operation of pilot-scale bioreactor was relatively stable, with influent nitrate concentrations in the range of 1–18 mg/L, and effluent nitrite and nitrate was below 0.5 mg/L and in the range of 0–6 mg/L (Figure 9). At the beginning of stable operation in winter, the lowest ambient temperature was −8 °C, and the lowest sewage temperature was 13 °C as the temperature decreased gradually. After 17 d, when the sewage temperature reached its lowest value, the effluent nitrate was 4.7 mg/L, resulting in the treatment efficiency at 57.7%, while still meeting the designed discharge standard. That is, temperature presented a substantial effect on the nitrate removal performance of sulfur-based autotrophic denitrification [11–13]. In addition, the influent nitrate in the plum-rain season decreased to 1–2 mg/L, and that of the effluent to 0 mg/L due to the low influent concentration. In summer, the sewage temperature increased to above 20 °C, and the nitrate removal increased to 85%. As depicted in Figure 9A, relatively high effluent nitrate could be observed as a consequence of untimely or incomplete backwashing, thus making the operation of pilot-scale bioreactor become poor. Moreover, based on the probability distribution of the effluent nitrate concentrations (Figure 9B), those lower than 4 mg/L accounted for approximately 87% of all concentration values, suggesting that the pilot-scale bioreactor was efficient in denitrification during long-term stable operation.

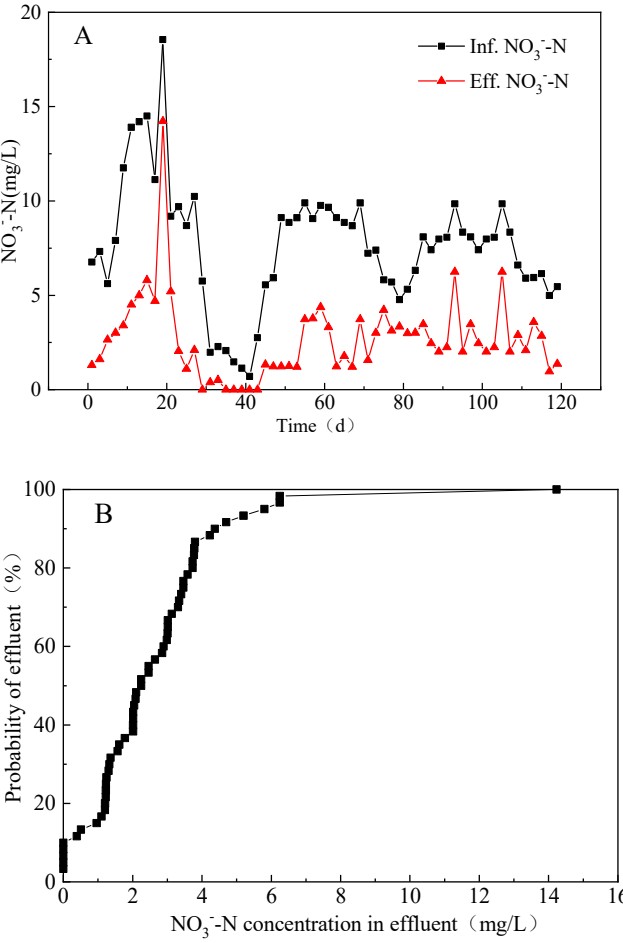

**Figure 9.** Nitrate removal (**A**) and effluent cumulative distribution (**B**) in the pilot-scale bioreactor.

### 3.4.2. Variation of Ammonia Nitrogen and pH Value

It can be seen in Figure 10 that the influent ammonia nitrogen of the pilot-scale bioreactor ranged from 0 to 1.7 mg/L, and that in the effluent ranged from 0 to 1.4 mg/L. There was essentially no significant change in the influent and effluent ammonia nitrogen concentrations because ammonia nitrogen was found to be not essential for nitrate removal and growth by sulfur-based autotrophic denitrification bacteria [14]. After 120 d, the effluent ammonia nitrogen was higher than that in the influent, primarily because untimely or incomplete backwashing caused the autolysis of detached biofilm in the anaerobic state [15] to release ammonia nitrogen. However, the effluent ammonia nitrogen still met the designed discharge standards.

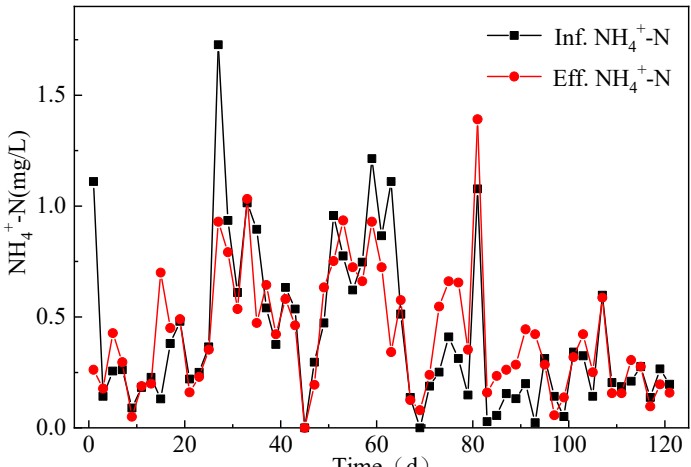

**Figure 10.** Ammonia nitrogen variation of pilot-scale bioreactor.

Sulfur autotrophic denitrifying bacteria will use alkalinity for nitrogen removal; therefore, the effluent pH decreased with an increasing level of nitrate removal (Figure 11). Nitrate removal can be effectively achieved without external alkalinity provided in the pilot-scale bioreactor. The influent pH of the pilot-scale bioreactor ranged from 6 to 7, with an average value of 6.93, whereas the effluent pH ranged from 6 to 6.9, with an average value of 6.39 [16]. This suggested that the pilot-scale bioreactor was able to effectively remove nitrate from the secondary effluent, and the effluent pH did not exceed the effluent discharge limit for wastewater treatment plants.

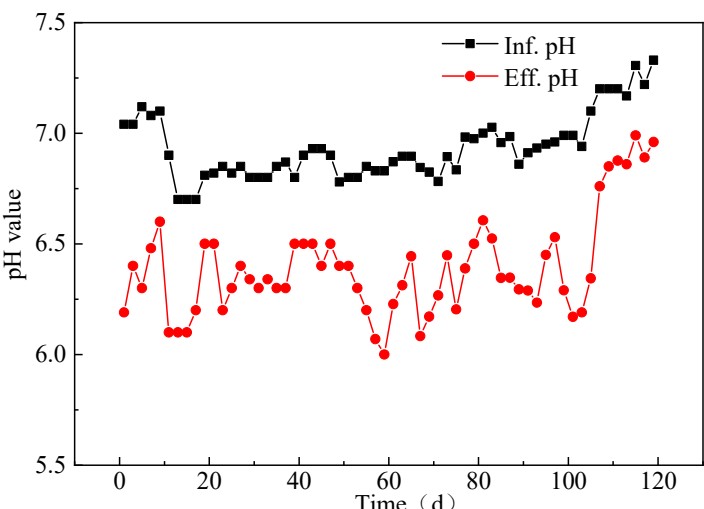

**Figure 11.** pH variation of pilot-scale bioreactor.

### 3.5. Backwash of the Pilot-Scale Bioreactor

Along with the long-term operation of the pilot-scale bioreactor, the biofilm gradually fell off and remained in the middle of the sulfur particle bed, and subsequently reduced the porosity of the sulfur particle bed and increased the water pressure inside the pilot-scale bioreactor. With the occurrence of short flow or sulfur particle floating, the treatment efficiency of the pilot-scale bioreactor significantly decreased. Therefore, timely and effective backwashing was essential to maintain efficient denitrification of the pilot-scale bioreactor, and it is also the main measure to discharge gaseous products in the pilot-scale bioreactor.

The pilot-scale bioreactor experienced an internal pressure rise during operation; however, nitrate removal performance will not be greatly influenced before the pressure reaches the critical value of 0.03 MPa. When the internal pressure exceeds the critical value, it would disperse the sulfur particle bed and trigger a short flow to relieve the pressure. Then, the flow rate increases and the HRT decreases, resulting in a serious decrease in the denitrification rate. Consequently, pressure is the basis for backwashing. In addition, effluent DO could be employed as an auxiliary basis. In the case of normal pressure at the bottom of the pilot-scale bioreactor, high effluent DO was observed, indicating that backwashing is required for the long-term stable operation of the pilot-scale bioreactor.

As illustrated in Figure 12, nitrate removal is about 50% before backwashing in the pilot-scale bioreactor, which increased to 80% within 30 min after backwashing. The aging biofilm could be effectively discharged by continuing backwashing for 2–3 min. Furthermore, the denitrification performance was able to recover to a normal level within 30 min after backwashing, and nitrate removal efficiency might be greatly improved. In general, the recovery time of the pilot-scale bioreactor after backwashing is relatively short, implying that the sulfur-based autotrophic denitrification process presents good applicability, feasibility and practicability.

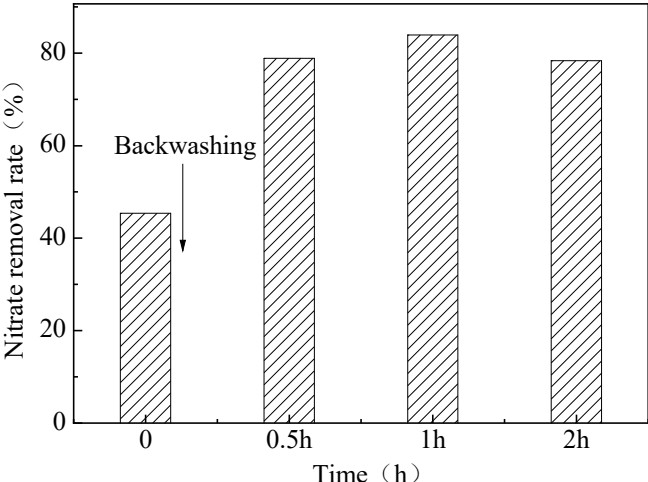

**Figure 12.** Backflush recovery of pilot-scale bioreactor.

### 3.6. Microbial Community Structure

3.6.1. Microbial Community Characteristics in the Lab-Scale and Pilot-Scale Bioreactor

The Shannon index was used to analyze the diversity of microbial populations on the sulfur particles, and the indices were 2.22 and 3.92 in the lab-scale and pilot-scale bioreactor, respectively (Figure 13). The microbial diversity in the pilot-scale bioreactor was significantly higher than that in the lab-scale bioreactor. The possible reason could be due to the complex sewage quality from the secondary sedimentation tank, presenting a greater biodiversity and higher impact resistance. On the contrary, the influent of the lab-scale bioreactor is simple and stable synthetic sewage, hence demonstrating a low biodiversity and smaller impact resistance.

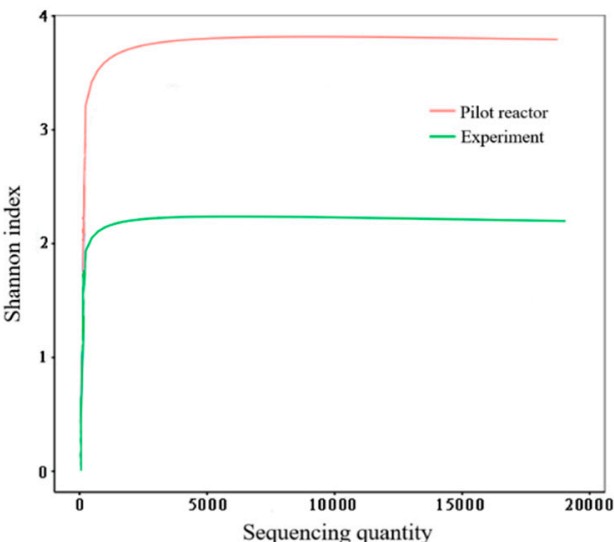

**Figure 13.** Sparse distribution of the Shannon index.

The microorganisms in the lab-scale bioreactor and pilot-scale bioreactor were categorized into more than seven phyla (Figure 14), where the principal phylum was dominated by Proteobacteria. The proportion of Proteobacteria in the lab-scale bioreactor is 94.5%, of which β-Proteobacteria accounted for 63.5%. However, 90.3% of Proteobacteria were accounted for in the pilot-scale bioreactor, 39.3% of which were β-Proteobacteria. Clearly, the percentage of β-Proteobacteria, an important phylum for the nitrogen removal [17], was significantly higher in the lab-scale bioreactor than in the pilot-scale bioreactor, indicating more diverse denitrifying functional microorganisms [8] in the sulfur particles enriched in the lab-scale bioreactor.

| Phylum | Pilot-scale bioreactor | Lab-scale bioreactor |
|---|---|---|
| *Proteobacteria* | 90.30 | 94.50 |
| *Bacteroidetes* | 1.97 | 4.10 |
| *Acidobacteria* | 1.81 | 0.07 |
| *Firmicutes* | 0.12 | 0.72 |
| *Planctomycetes* | 0.79 | 0.03 |
| *Chloroflexi* | 0.68 | 0.02 |
| *Ignavibacteriae* | 0.55 | 0.11 |
| *others* | 3.78 | 0.44 |

**Figure 14.** Microbial community structure at the phylum level in the lab-scale and pilot-scale bioreactor.

Microbial community structure at genus level in lab-scale and pilot-scale bioreactor was depicted in Figure 15, *Thiobacillus* sp., *Sulfurimonas* sp. and *Sulfuricella* sp. were the main functional microorganisms, accounting for 37.61% and 18.0%, 20.39% and 0.93%, and 0.73% and 7.67%, respectively. *Thiobacillus* sp., a typical functional strain in sulfur-based autotrophic denitrification, performs good nitrate removal performance by using reduced sulfur, [18], whilst *Sulfurimonas* sp. and *Sulfuricella* sp. can oxidize sulfur for achieving high denitrification properties [19,20]. In addition, functional microorganisms

on the sulfur particle bed of the lab-scale bioreactor had a relative abundance of 58.7%, while their abundance was merely 26.6% for the pilot-scale bioreactor, which was consistent with the discrepancy in the maximum denitrification load between the lab-scale and pilot-scale bioreactor.

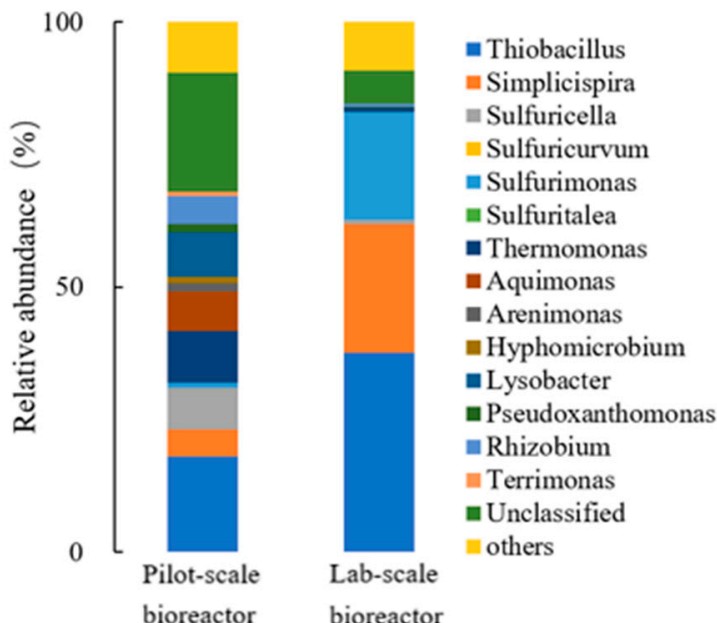

**Figure 15.** Microbial community structure at the genus level in the lab-scale and pilot-scale bioreactor.

### 3.6.2. Microbial Community Characteristics in the Sulfur Particle Bed

To clarify the flora succession of the microbial community on the sulfur particle bed at different heights, the upper (at 1.6–1.8 m), middle (at 1.1–1.3 m), and lower (at 0.6–0.8 m) sulfur particle bed of the pilot-scale bioreactor were sampled to be analyzed (Figure 16).

| Phylum | S-upper part | S-middle part | S-lower part |
|---|---|---|---|
| *Proteobacteria* | 54.9 | 68.2 | 51.0 |
| *Bacteroidetes* | 19.4 | 7.3 | 19.2 |
| *Firmicutes* | 5.8 | 8.9 | 5.3 |
| *Chloroflexi* | 1.8 | 3.5 | 5.7 |
| *Spirochaetes* | 6.9 | 3.1 | 4.6 |
| *Chlorobi* | 1.8 | 0.9 | 1.7 |
| *Actinobacteria* | 1.0 | 1.5 | 3.0 |
| *Planctomycetes* | 1.0 | 0.9 | 1.7 |
| *Acidobacteria* | 1.5 | 1.0 | 1.0 |
| *Others* | 4.8 | 3.3 | 4.3 |
| *Unknown* | 0.9 | 1.2 | 2.4 |

**Figure 16.** Microbial community structure at the phylum level at different heights in the pilot-scale bioreactor.

The microbial flora in the pilot-scale bioreactor contained nine types of bacterial phyla, namely *Proteobacteria*, *Bacteroidetes*, *Firmicutes*, *Spirochaetes*, *Chloroflexi*, *Chlorobi*, *Actinobacteria*, *Planctomycetes*, and *Acidobacteria*. *Proteobacteria* are dominant in the upper, middle and lower parts of biofilms and play an important role in nitrogen removal. Many confirmed denitrification bacteria belong to *Proteobacteria*. The proportion of *Proteobacteria* in the upper, middle and lower parts of the reactor is 54.9%, 68.2% and 51%, respectively, which is consistent with the community structure of the sulfur autotrophic denitrification system [21–23] and WWTPs [24]. It mainly consisted of *β-Proteobacteria*, which had a slightly higher proportion in the middle part of the sulfur particle bed than in the upper and lower parts, indicating that the middle part provided a more suitable nitrate removal environment for the growth of denitrifying microorganisms. In addition to *Proteobacteria*, *Bacteroidetes* were the most abundant group, accounting for 7.3%~19.4% of all sludge samples.

The microflora at different heights of the sulfur particle bed were further divided by genus into the main functional microorganisms *Thiobacillus* sp., *Sulfurimonas* sp., and *Sulfuricurvum* sp. (Figure 17). *Thiobacillus* sp. and *Sulfurimonas* sp. are autotrophic denitrifying bacteria, and the reduction of nitrate is mainly obtained through the denitrification process of *Thiobacillus* sp. and *Sulfurimonas* sp. As obligate chemoautotrophic bacteria, *Thiobacillus* sp. is the main denitrification bacteria in the process of sulfur autotrophic denitrification process, and is also a typical sulfur oxidizing denitrification bacteria. *Thiobacillus* sp. reduced sulfides to sulfate and gain energy, and nitrate can be reduced along with the oxidation of sulfides [25]. In addition, *Thiobacillus* sp. was dominant in the upper, middle, and lower parts, and the increasing relative abundance of *Thiobacillus* sp. was consistent with the trend of relative abundance of *Proteobacteria*, the proportion of the middle part is significantly higher than that of the upper and lower parts, and the proportion of the middle part is 15.7%. *Thiobacillus* sp. ratio in the upper sulfur particle bed decreased probably due to the gradual decrease of effluent nitrate. *Sulfurimonas* sp. is a facultative anaerobic autotrophic bacterium capable of denitrification by using sulfur and sulfide as electron donors [26], and it was more abundant in the upper part of the pilot-scale bioreactor (approximately 10%). *Sulfuricurvum* sp. is a type of facultative anaerobic microorganism, which is able to use sulfide, elemental sulfur and thiosulfate as electron donor to achieve autotrophic denitrification [27–29]. In addition, the proportion of *Sulfuricurum* sp. in the upper, middle and lower levels presents little difference. The relative abundance of microbial populations in the sulfur particle bed at different heights correlated well with the variations of DO and nitrate concentration. Moreover, the sulfur particles at the bottom of the denitrification layer participated in DO removal when the denitrification biofilm could not completely remove the influent DO, resulting in a lower relative abundance of denitrifying microorganisms at the bottom of the sulfur particle bed. However, it ensures a suitable nitrate removal environment in the middle part of the sulfur particle bed. Therefore, the relative abundance of denitrifying functional microorganisms in the middle part is higher than that in the bottom of the sulfur particle bed.

*3.7. Operation Cost of Pilot-Scale Bioreactor*

The operation costs of the pilot-scale bioreactor included sulfur particles consumption and bioreactor electricity consumption, and the operation period comprised the commissioning operation and stable operation period. The commissioning operation period lasted for 120 days, during which 21400 $m^3$ of wastewater was processed, with the sulfur particles consumption of approximately 487 kg and nitrogen removal of 156.7 kg. In addition, the stable operation period lasted for 109 d, during which 42,670 $m^3$ of wastewater was processed, with the sulfur particles consumption of 674 kg and nitrogen removal of 216 kg. Therefore, the processed wastewater volume was up to 64070 $m^3$, with approximately 1161 kg of sulfur particles consumed and 372.7 kg of nitrogen removed during the operation period, resulting in an actual N:S ratio of 1:3.11. However, according to the characteristics of the sulfur-based autotrophic denitrification bacteria, the N:S ratio was theoretically 1:2.51. A total of 3.11 g of sulfur particles was consumed to remove 1 g of nitrate during practical

operation because part of the sulfur particles likely floated up and was lost with the effluent when backwashing was insufficient. Moreover, the sulfur particles progressively became smaller along with the denitrification process and subsequently during backwashing. In addition, the decrease of influent DO could consume sulfur particles, which synergistically led to a high N:S ratio.

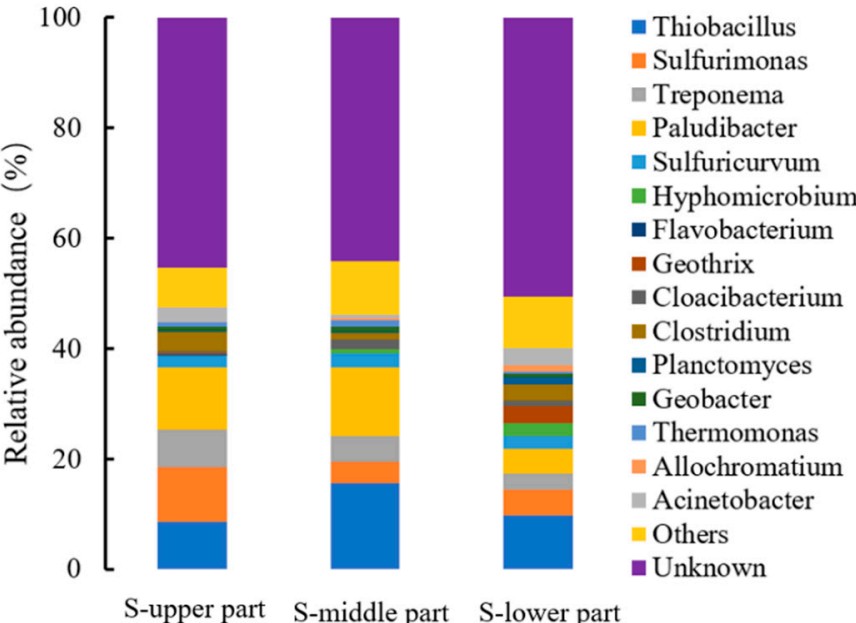

**Figure 17.** Microbial community structure at genus level at different heights in pilot-scale bioreactor.

(1)    Sulfur particles consumption

The sulfur particles are industrial grade with a market price of 215.60 $/t. Approximately 1161 kg of sulfur particles was consumed with the sulfur loss of 250.32 $ to process 64,070 m$^3$ of wastewater; thus, the cost of sulfur particles consumed was 0.004 $/t.

$$\text{Sulfur particles consumption} = \frac{215.60 \times 1.161}{64,070} \approx 0.004 \ \$/t$$

(2)    Bioreactor electricity consumption

The lifting pump was a vertical centrifugal pump with a power of 1.1 kW, which operated for 229 d in total. Since the backwash pump operated infrequently and only for a short period of time, its electricity consumption costs were negligible. Therefore, the electricity consumption (electricity price subject to the large industrial electricity price of 0.093 $/kW·h) was 0.009 $/t.

$$\text{Bioreactor electricity consumption} = \frac{1.1 \times 24 \times 229 \times 0.093}{64,070} \approx 0.009 \ \$/t$$

Consequently, the operation cost of the pilot-scale bioreactor was 0.009 $/t. Moreover, the cost of the sulfur carrier loss from the pilot-scale bioreactor to remove 5 mg/L of nitrate was 0.003 $/t, compared with the sodium acetate cost of 0.022 $/t (the market price of sodium acetate is 409.66 $/t) to remove 5 mg/L nitrate during heterotrophic denitrification, where acetic acid or sodium acetate are used as the carbon source. This indicates that sulfur-based autotrophic denitrification involves significantly lower sulfur particles consumption costs than the addition of organic carbon, which may cause incomplete denitrification or secondary pollution in the effluent due to an insufficient or excessive dosage.

## 4. Conclusions

The pilot-scale bioreactor was started in a low-flow continuous influent and effluent mode and operated for approximately 120 d. It was found that the optimal HRT was 0.21 h, with which the maximum denitrification load reached 1158 mg/(L·d), and the denitrification rate was 164 $gNO_3{}^-$-N//(m$^3$·h). During the long-term stable operation period, the proportion of effluent nitrate lower than 4 mg/L was 87%, implying that the treatment performance is relatively stable, and the effluent pH and ammonia nitrogen are superior to the discharge standard. Effective backwashing maintained an efficient denitrification performance of the pilot-scale bioreactor, and the denitrification performance could be recovered within 0.5 h after backwashing. Therefore, the pressure variation at the bottom of the pilot-scale bioreactor could be used as the basis for backwashing. In addition, no additional alkalinity was provided during the operation process, resulting in less stringent operating conditions. The *Thiobacillus* sp. ratio in the upper sulfur particle bed decreased probably due to the gradual decrease of effluent nitrate. The complex influent quality also led to a discrepancy in the microbial community structure, and other denitrifying microorganisms accounted for a certain proportion, which also caused the reduction of *Thiobacillus* abundance. Furthermore, the operation cost is merely 0.013 $/t, indicating the great economic applicability of the sulfur-based autotrophic denitrification process.

**Author Contributions:** J.L.: Project administration; Y.W. and W.X.: Roles/Writing—original draft; F.Q. and J.L.: Writing—review and editing; Y.W., W.X., X.Y., Z.R. and K.H.: Investigation and Methodology. All authors listed have made a substantial, direct and intellectual contribution to the work and approved the final manuscript for publication. All authors have read and agreed to the published version of the manuscript.

**Funding:** The authors gratefully acknowledge the financial support provided by Jiangsu Policy Guidance Program (International Science and Technology Collaboration) (BZ2021030), Wuxi Innovation and Entrepreneurship Program for Science and Technology (M20211003), and the Pre-research Fund of Jiangsu Collaborative Innovation Center of Technology and Material of Water Treatment (XTCXSZ2020-2).

**Data Availability Statement:** Not applicable.

**Conflicts of Interest:** All the authors listed have approved the manuscript and agreed to authorship and submission of the manuscript for peer review.

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
