# Peer review of "Long-Term Operation of a Pilot-Scale Sulfur-Based Autotrophic Denitrification System for Deep Nitrogen Removal"

_water, doi:10.3390/w15030428_

Round 1
Reviewer 1 Report
Carbon source addition has always been an important factor in increasing the operation cost of WWTPs. This paper studies the sulfur autotrophic denitrification pilot-scale denitrification system, which effectively eliminate the problem of carbon source addition and ensure the stable denitrification effect of WWTPs. The start-up, stable operation, microbial community characteristics and operation cost of the pilot-scale bioreactor were analyzed in detail. This paper is well developed, and can be published after minor revision.
1. L66, why does denitrification process involve ammonia nitrogen?
2. Section 2.1, the operation parameter information of lab-scale reactor needs to be supplemented.
3. L131-L136, the description is not clear enough, it is recommended to rewrite.
4. Does the low temperature stage affect the backwashing performance?
5. Section 3.6.2, it is recommended to supplement the characteristics of other major functional strains.
6. L385-L387, the operation conclusion of the lab-scale reactor is irrelevant to this study, and the conclusion of the microbial community structure characteristics of the pilot- scale bioreactor needs to be supplemented.
Author Response
Dear Reviewer #1:
Thank you for your letter and the comments concerning our manuscript entitled “Long-term operation of pilot-scale sulfur-based autotrophic denitrification system for deep nitrogen removal”. (Manuscript ID water-2148058).
The comments are all valuable and very helpful for revising and improving our paper, as well as the important guiding significance to our researches. We have studied comments carefully and have made corrections which we hope meet with approval. Revised portion are also marking with yellow in the revised paper. The main corrections in the paper and the responds to the comments of reviewer #1 are as following:
Responds to the reviewer’s comments:
Carbon source addition has always been an important factor in increasing the operation cost of WWTPs. This paper studies the sulfur autotrophic denitrification pilot-scale denitrification system, which effectively eliminate the problem of carbon source addition and ensure the stable denitrification effect of WWTPs. The start-up, stable operation, microbial community characteristics and operation cost of the pilot-scale bioreactor were analyzed in detail. This paper is well developed, and can be published after minor revision.
- L66, why does denitrification process involve ammonia nitrogen?
Response: After verification, it was confirmed that a small amount of CO2 and ammonia nitrogen would be used by Thiobacillus sp. for denitrification.
- Section 2.1, the operation parameter information of lab-scale reactor needs to be supplemented.
Response: The basic information of lab-scale bioreactor was provided in Section 2.1 as:
In addition, a lab-scale bioreactor was established to simultaneously evaluate the nitrogen removal performance of the pilot-scale bioreactor, with the same operation parameters. The effective volume was 8.0 L (inner diameter: 20cm; height: 35 cm), and the coarse sand and sulfur particles was filled in the bioreactor.
- L131-L136, the description is not clear enough, it is recommended to rewrite.
Response: The sentence had been revised as “Stage II (31-130 d), sludge inoculation was terminated and the influent nitrate was maintained up to 30 mg/L by adding KNO3, which is beneficial to the growth of autotrophic denitrification biofilm. Moreover, the pilot-scale bioreactor ought to be periodically and completely discharged to prevent excessive accumulation of toxic substances from microorganisms, whilst the influent flow rate was kept at 1 m3/h.”
- Does the low temperature stage affect the backwashing performance?
Response: According to the nitrogen removal performance of pilot-scale bioreactor during low temperature period, low temperature stage will not affect the backwashing performance.
- Section 3.6.2, it is recommended to supplement the characteristics of other major functional strains.
Response: Section 3.6.2 had been revised and more references had been added in the manuscript.
3.6.2 Microbial community characteristics in the sulfur particle bed
To clarify the flora succession of microbial community on the sulfur particle bed at different heights, the upper (at 1.6-1.8 m), middle (at 1.1-1.3 m), and lower (at 0.6-0.8 m) sulfur particle bed of the pilot-scale bioreactor were sampled to analyzed (Figure 16).
The microbial flora in the pilot-scale bioreactor contained nine type of bacterial phyla, namely Proteobacteria, Bacteroidetes, Firmicutes, Spirochaetes, Chloroflexi, Chlorobi, Actinobacteria, Planctomycetes, and Acidobacteria. Proteobacteria are dominant in the upper, middle and lower parts of biofilms and play an important role in nitrogen removal. Many confirmed denitrification bacteria belong to Proteobacteria. The proportion of Proteobacteria in the upper, middle and lower parts of the reactor is 54.9%, 68.2% and 51%, respectively, which is consistent with the community structure of sulfur autotrophic denitrification system[21-23] and WWTPs[24]. It was mainly consisted of β-Proteobacteria, which had a slightly higher proportion in the middle part of the sulfur particle bed than in the upper and lower parts, indicating that the middle part provided a more suitable nitrate removal environment for the growth of denitrifying microorganisms. In addition to Proteobacteria, Bacteroidetes were the most abundant group, accounting for 7.3% ~ 19.4% of all sludge samples.
The microflora at different heights of the sulfur particle bed was further divided by genus into the main functional microorganisms Thiobacillus sp., Sulfurimonas sp., and Sulfuricurvum sp (Figure 17). Thiobacillus sp. and Sulfurimonas sp. are autotrophic denitrifying bacteria, and the reduction of nitrate is mainly obtained through the denitrification process of Thiobacillus sp. and Sulfurimonas sp. As an obligate chemoautotrophic bacteria, Thiobacillus sp. is the main denitrification bacteria in the process of sulfur autotrophic denitrification process, and is also a typical sulfur oxidizing denitrification bacteria. Thiobacillus sp. reduced sulfides to sulfate and gain energy, and nitrate can be reduced along with the oxidation of sulfides[25]. In addition, Thiobacillus sp. was dominant in the upper, middle, and lower parts, and the increasing relative abundance of Thiobacillus sp. was consistent with the trend of relative abundance of Proteobacteria, the proportion of the middle part is significantly higher than that of the upper and lower parts, and the proportion of the middle part is 15.7%. Thiobacillus sp. ratio in the upper sulfur particle bed decreased probably due to the gradual decrease of effluent nitrate. Sulfurimonas sp. is a facultative anaerobic autotrophic bacterium capable of denitrification by using sulfur and sulfide as electron donors[26], and it was more abundant in the upper part of the pilot-scale bioreactor (approximately 10%). Sulfuricurvum sp. is a type of facultative anaerobic microorganism, which is able to use sulfide, elemental sulfur and thiosulfate as electron donor to achieve autotrophic denitrification[27-29]. In addition, the proportion of Sulfuricurum sp. in the upper, middle and lower levels presents little difference. The relative abundance of microbial populations in the sulfur particle bed at different heights correlated well with the variations of DO and nitrate concentration. Moreover, the sulfur particles at the bottom of the denitrification layer participated in DO removal when the denitrification biofilm could not completely remove the influent DO, resulting in a lower relative abundance of denitrifying microorganisms at the bottom of the sulfur particle bed. However, it ensures a suitable nitrate removal environment in the middle part of the sulfur particle bed. Therefore, the relative abundance of denitrifying functional microorganisms in the middle part is higher than that in the bottom of the sulfur particle bed.
- Zhang, L.; Zhang, C.; Hu, C.; Liu, H.; Qu, J. Denitrification of groundwater using a sulfur-oxidizing autotrophic denitrifying anaerobic fluidized-bed MBR: performance and bacterial community structure. Applied Microbiology & Biotechnology 2015, 99(6):2815.
- Morgan-Sagastume, F.; Nielsen, J.L.; Nielsen, P.H. Substrate-dependent denitrification of abundant probe-defined denitrifying bacteria in activated sludge. FEMS Microbiology Ecology 2008, 66(2):447-461.
- Zhang, T.; Shao, M.; Ye, L. 454 Pyrosequencing reveals bacterial diversity of activated sludge from 14 sewage treatment plants. Isme Journal 2012, 6(6):1137-1147.
- Wang, X.; Wen, X.; Yan, H.; Ding, K.; Zhao, F.; Hu, M. Bacterial community dynamics in a functionally stable pilot-scale wastewater treatment plant. Bioresour Technol 2011, 102(3):2352-2357.
- Shao, M.F.; Zhang, T.; Fang, H.P. Sulfur-driven autotrophic denitrification: Diversity, biochemistry, and engineering applications. Applied Microbiology and Biotechnology 2010, 88(5):1027-1042.
- Pang, Y.; Wang, J. Various electron donors for biological nitrate removal: A review. Science of The Total Environment 2021, 794(20):148699.
- Erkan; Sahinkaya; Nesrin; Dursun Use of elemental sulfur and thiosulfate as electron sources for water denitrification. Bioprocess & Biosystems Engineering 2015.
- Tang, B.; Xiang, Q.; Wang, J.; Zhang, Y.; Li, X.; Cheng, H.; Hu, M.; Zou, Z. Kinetics of limestone decomposition in hot metal. Metallurgical Research and Technology 2018, 115(6).
- C, S.W.A.B.; A, Y.K.Y.; A, X.G.C.; C, J.Z.L.A.B.; C, J.L.A.B. Effects of bamboo powder and rice husk powder conditioners on sludge dewatering and filtrate quality. International Biodeterioration & Biodegradation 2017, 124:288-296.
- L385-L387, the operation conclusion of the lab-scale reactor is irrelevant to this study, and the conclusion of the microbial community structure characteristics of the pilot- scale bioreactor needs to be supplemented.
Response: The Conclusion had been revised as:
The pilot-scale bioreactor was started in a low-flow continuous influent and effluent mode and operated for approximately 120 d. It was found that the optimal HRT was 0.21 h, with which the maximum denitrification load reached 1158 mg/(L·d), and the denitrification rate was 164 gNO3--N//(m3·h). During the long-term stable operation period, the proportion of effluent nitrate lower than 4 mg/L is 87%, implying that the treatment performance is relatively stable, and the effluent pH and ammonia nitrogen are superior to the discharge standard. Effective backwashing maintained an efficient denitrification performance of the pilot-scale bioreactor, and the denitrification performance can be recovered within 0.5 h after backwashing. Therefore, the pressure variation at the bottom of the pilot-scale bioreactor could be used as the basis for backwashing. In addition, no additional alkalinity was provided during the operation process, resulting in less stringent operating conditions. Thiobacillus sp. ratio in the upper sulfur particle bed decreased probably due to the gradual decrease of effluent nitrate. The complex influent quality also led to discrepancy in microbial community structure, and other denitrifying microorganisms account for a certain proportion, which also cause the reduction of Thiobacillus abundance. Furthermore, the operation cost is merely 0.013 $/t, indicating the great economic applicability of the sulfur-based autotrophic denitrification process.
In addition, the language and format of the paper had both been revised and marked in yellow to achieve the publication of this manuscript.
We tried our best to improve the manuscript and made some changes in the manuscript. These changes will not influence the content and framework of the paper. We appreciate for your warm work earnestly, and hope that the correction will meet with approval.
Special thanks to you for your good comments.
Kind Regards
Yan WANG and Ji LI
Reviewer 2 Report
This is a high-quality article, which has carried out a detailed study on the start-up, long-term stable operation, backwashing strategy formulation, microbial community structure analysis and cost accounting of the pilot-scale bioreactor. Although the sulfur autotrophic denitrification system is only built on the basis of pilot-scale bioreactor, the author's research is more in-depth and comprehensive, and the data reliability is high, which is expected to provide support for the practical application of low carbon emission denitrification technology. I suggest that the paper can be published after the following contents are provided.
1. The basic information of lab-scale bioreactor is not described in Chapter 2, which leads the reviewer to have some doubts when analyzing the microbial community structure.
2. Figure 5 can be used as a supplementary figure.
Author Response
Dear Reviewer #2:
Thank you for your letter and the comments concerning our manuscript entitled “Long-term operation of pilot-scale sulfur-based autotrophic denitrification system for deep nitrogen removal”. (Manuscript ID water-2148058).
The comments are all valuable and very helpful for revising and improving our paper, as well as the important guiding significance to our researches. We have studied comments carefully and have made corrections which we hope meet with approval. Revised portion are also marking with yellow in the revised paper. The main corrections in the paper and the responds to the comments of reviewer #2 are as following:
Responds to the reviewer’s comments:
This is a high-quality article, which has carried out a detailed study on the start-up, long-term stable operation, backwashing strategy formulation, microbial community structure analysis and cost accounting of the pilot-scale bioreactor. Although the sulfur autotrophic denitrification system is only built on the basis of pilot-scale bioreactor, the author's research is more in-depth and comprehensive, and the data reliability is high, which is expected to provide support for the practical application of low carbon emission denitrification technology. I suggest that the paper can be published after the following contents are provided.
- The basic information of lab-scale bioreactor is not described in Chapter 2, which leads the reviewer to have some doubts when analyzing the microbial community structure.
Response: The basic information of lab-scale bioreactor was provided as:
In addition, a lab-scale bioreactor was established to simultaneously evaluate the nitrogen removal performance of the pilot-scale bioreactor, with the same operation parameters. The effective volume was 8.0 L (inner diameter: 20cm; height: 35 cm), and the coarse sand and sulfur particles was filled in the bioreactor.
Furthermore, more information for the microbial community structure in both lab-scale and pilot-scale bioreactor was analyzed in-depth.
- Figure 5 can be used as a supplementary figure.
Response: The generation time of autotrophic bacteria is longer. In order to prove the effect of biofilm formation, the author believes that Figure 5 is more suitable as a normal figure.
In addition, the language and format of the paper had both been revised and marked in yellow to achieve the publication of this manuscript.
We tried our best to improve the manuscript and made some changes in the manuscript. These changes will not influence the content and framework of the paper. We appreciate for your warm work earnestly, and hope that the correction will meet with approval.
Special thanks to you for your good comments.
Kind Regards
Yan WANG and Ji LI
Reviewer 3 Report
This manuscript discusses the properties of pilot-scale sulfur autotrophic denitrification bioreactor, which excavates new idea from the perspective of long-term stable operation, microbial community structure and operational cost. It further reveals the potential of low-carbon emission denitrification process, which is a very promising study for practical application. However, this manuscript need to minor revise before accepted.
1. In this study, the authors should provide the operational parameters and nitrate removal performance of the lab-scale bioreactor, which is not mentioned in the whole manuscript.
2. L139 and L360, why does the calculation of bioreactor electricity consumption not include the energy consumption of heater during low temperature operation of pilot-scale bioreactor?
3. L144,35-40 mg/(L-d) should be 35-40 mg/(L•d).
4. L154 and L160, what is the difference between nitrogen load and denitrification load?
5. L188-L190, this sentence needs to be rewritten.
6. Section 3.4, have the authors monitored the effluent COD from the secondary sedimentation tank and the pilot-scale bioreactor, and dose the heterotrophic denitrification occurred in the pilot-scale bioreactor?
Author Response
Dear Reviewer #3:
Thank you for your letter and the comments concerning our manuscript entitled “Long-term operation of pilot-scale sulfur-based autotrophic denitrification system for deep nitrogen removal”. (Manuscript ID water-2148058).
The comments are all valuable and very helpful for revising and improving our paper, as well as the important guiding significance to our researches. We have studied comments carefully and have made corrections which we hope meet with approval. Revised portion are also marking with yellow in the revised paper. The main corrections in the paper and the responds to the comments of reviewer #3 are as following:
Responds to the reviewer’s comments:
This manuscript discusses the properties of pilot-scale sulfur autotrophic denitrification bioreactor, which excavates new idea from the perspective of long-term stable operation, microbial community structure and operational cost. It further reveals the potential of low-carbon emission denitrification process, which is a very promising study for practical application. However, this manuscript need to minor revise before accepted.
- In this study, the authors should provide the operational parameters and nitrate removal performance of the lab-scale bioreactor, which is not mentioned in the whole manuscript.
Response: The basic information of lab-scale bioreactor was provided in Section 2.1 as:
In addition, a lab-scale bioreactor was established to simultaneously evaluate the nitrogen removal performance of the pilot-scale bioreactor, with the same operation parameters. The effective volume was 8.0 L (inner diameter: 20cm; height: 35 cm), and the coarse sand and sulfur particles was filled in the bioreactor.
- L139 and L360, why does the calculation of bioreactor electricity consumption not include the energy consumption of heater during low temperature operation of pilot-scale bioreactor?
Response: The heater only operates for about 15 days, and its power is only 20% of the practical equipment. Therefore, the energy consumption is not included in the electricity consumption.
- L144,35-40 mg/(L-d) should be 35-40 mg/(L•d).
Response: The unit had been revised as “mg/(L•d)”.
- L154 and L160, what is the difference between nitrogen load and denitrification load?
Response: The “nitrogen load” in the manuscript had been changed to “denitrification load”.
- L188-L190, this sentence needs to be rewritten.
Response: The sentence had been revised as “Therefore, in order to achieve high nitrate removal performance, it is necessary to remove DO of the secondary effluent and thus create good denitrification environment for autotrophic denitrifying bacteria.”
- Section 3.4, have the authors monitored the effluent COD from the secondary sedimentation tank and the pilot-scale bioreactor, and dose the heterotrophic denitrification occurred in the pilot-scale bioreactor?
Response: During about five-months operation of the pilot-scale bioreactor, the average effluent COD of the secondary sedimentation tank from WWTP is about 40 mg/L, and the average effluent COD of pilot-scale bioreactor is merely 20 mg/L. The effluent COD after biochemical treatment is difficult to further degrade, thus there will be no heterotrophic denitrification phenomenon in the pilot-scale bioreactor.
In addition, the language and format of the paper had both been revised and marked in yellow to achieve the publication of this manuscript.
We tried our best to improve the manuscript and made some changes in the manuscript. These changes will not influence the content and framework of the paper. We appreciate for your warm work earnestly, and hope that the correction will meet with approval.
Special thanks to you for your good comments.
Kind Regards
Yan WANG and Ji LI
Reviewer 4 Report
-The theme of the work presented is current and relevant.
-The experimental design is clear and appropriate for achieving the desired results.
The workload is high and is appropriate to the results presented.
Author Response
Dear Reviewer #4:
Thank you for your letter and the comments concerning our manuscript entitled “Long-term operation of pilot-scale sulfur-based autotrophic denitrification system for deep nitrogen removal”. (Manuscript ID water-2148058).
The comments are all valuable and very helpful for revising and improving our paper, as well as the important guiding significance to our researches. We have studied comments carefully and have made corrections which we hope meet with approval. Revised portion are also marking with yellow in the revised paper. The main corrections in the paper and the responds to the comments of reviewer #4 are as following:
Responds to the reviewer’s comments:
The theme of the work presented is current and relevant. The experimental design is clear and appropriate for achieving the desired results. The workload is high and is appropriate to the results presented.
Response: The manuscript had been carefully revised.
In addition, the language and format of the paper had both been revised and marked in yellow to achieve the publication of this manuscript.
We tried our best to improve the manuscript and made some changes in the manuscript. These changes will not influence the content and framework of the paper. We appreciate for your warm work earnestly, and hope that the correction will meet with approval.
Special thanks to you for your good comments.
Kind Regards
Yan WANG and Ji LI